# Structural, Familial, and Psychosocial Factors Affecting Long-Term Antiretroviral Treatment Adherence amongst Adolescents Living with Perinatally Acquired HIV in Limpopo, South Africa

**DOI:** 10.3390/ijerph20227074

**Published:** 2023-11-17

**Authors:** Rirhandzu Austice Mabasa, Sphiwe Madiba, Tebogo Maria Mothiba

**Affiliations:** Faculty of Health Sciences, Executive Deans Office, University of Limpopo, Polokwane 0700, South Africa; sphiwe.madiba@ul.ac.za (S.M.); tebogo.mothiba@ul.ac.za (T.M.M.)

**Keywords:** adolescents, adherence, lifelong ART, perinatal HIV, rural South Africa, structural factors

## Abstract

After more than two decades of the expansion of antiretroviral treatment (ART) in adolescents living with perinatal HIV (APHIV) in sub-Saharan Africa, there is still poorly sustained long-term adherence to ART due to multifactorial factors with the consequence of increased mortality and morbidity. There are little data available on the familial and structural factors which affect sustenance to long-term adherence to ART. A qualitative exploratory design was used to conduct in-depth interviews with 21 APHIV attending HIV care and management in the rural health facilities of Vhembe district in Limpopo Province, South Africa. Transcripts were translated verbatim into English, and data were analyzed using Tesch’s eight steps of qualitative data analysis. The sample consisted of APHIV 10–19 years old who were aware of their HIV status, and all had received ART for more than 5 years. They lived in extended, disrupted, grandparent- and child-headed households. They experienced food insecurities due to poverty or orphanhood, as well as living in disrupted households, which deterred them from long-term adherence. In addition, dependency on social support grants to sustain their livelihoods affected long-term adherence. APHIV had challenges with structural factors such as inconsistent clinic attendance, clashes between school activities and clinic appointments, and the lack of transport fare to the clinic, which affected adherence. Although APHIV were on one-pill fixed-dose ART, they were not able to sustain long-term adherence due to various familial, structural, and psychosocial challenges. In addition to institution-based interventions, there is a need for family, community-based, and multi-sectorial interventions to support long-term ART adherence among APHIV.

## 1. Introduction

The expansion of antiretroviral therapy (ART) over the past two decades has seen an improvement in the disease progression and the lives of many people living with HIV. The effect of this improvement has been reduced mortality and morbidity, as well as viral load suppression [1]. However, the same level of improvement in mortality and morbidity has not been observed amongst adolescents living with perinatal HIV infection (APHIV) [2]. There is evidence that, compared to adults and children, adolescents fail to cope with living with HIV, which is evidenced by a variety of factors such as a high rate of virological failure and loss at follow-up, as well as high defaulter rates [2,3,4,5]. United Nations Children’s Fund’s (UNICEF) data indicated an increase of about 50% in AIDS-related deaths among APHIV, while they observed mortality decrease among all other age groups [6]. 

Currently, there are more than 1.7 million adolescents between the ages of 10 and 19 years living with HIV globally [7,8,9]. Adolescents aged 10–19 years make up a growing proportion of people living with HIV around the world [6,7,8]. About 86% of APHIV are from sub-Saharan Africa (SSA), which includes South Africa, and only 59% of them are receiving ART [7].

The introduction of effective ART changed the natural history of HIV infection, allowing an essentially normal life expectancy for those who can take and are adherent to therapy, both in developed and developing countries [3]. In South Africa, the majority of APHIV are currently on a one-pill, once-daily, fixed-dose combination antiretroviral regimen (Tenofovir, Emitricibine, and Efavirenz). In South Africa, this regimen was first approved in 2013 [4,7,10]. Adherence to ART remains low among APHIV in SSA despite multiple interventions implemented to improve ART adherence [4]. However, adolescents struggle with sustaining adherence over time, which is a necessity for the effectiveness of ART [11,12,13]. A systematic review reported an ART adherence rate of 65% among APHIV in the SSA region [4]. Qualitative data from studies conducted in SSA found that only 37% of APHIV were reported to be adherent [14,15]. According to the World Health Organization (WHO), sustaining adherence means attending 80% of clinic appointments to collect medication and 85% viral load suppression [14].

Much of the research on barriers to ART adherence among APHIV have focused on patient, health service, and medication-related factors [5,16]. Factors such as transitioning from childhood to adolescence, which affect adherence to treatment, have also been a focus of research [15,16]. At the same time, other studies have established the psychosocial and mental health challenges that indirectly contribute to APHIV failing to sustain long-term adherence to ART [17,18]. However, there is limited literature on the structural and familial factors that affect long-term adherence to ART in APHIV and how poor adherence compromises ART effectiveness [19]. While research shows that family plays a vital protective role and provides financial and emotional support to achieve optimal ART adherence [20], how the family influences adherence is not well established in SSA. 

Individual, social, and structural challenges that APHIV face in their daily lives create substantial barriers to retention in care and adherence to ART [21]. As stated, limited data from SSA show that APHIV are faced with social, cultural, and structural challenges, such as poverty and food insecurity, which affect their capacity to sustain long-term adherence to ART [12,22,23]. In many low-income and middle-income countries (LMICs), food insecurity has emerged as a major barrier to adherence and treatment for people living with HIV and AIDS (PLHIV) [24,25]. However, most studies focused on associations between food insecurity and adherence to ART among the general adult HIV-infected population [24,25]. Food insecurity is defined as when people are not able to access sufficiently safe and nutritious food for normal growth and development and for an active and healthy life [26].

There is notable recorded progress on the joint United Nations Programme on HIV/AIDS’ 90-90-90 approach that envisioned that 90% of HIV-positive people being treated, 90% of those being enrolled in ART, and 90% of those in ART achieving virological suppression by 2030 [27]. UNICEF has designed strategies such as peer-based group interventions and adolescent-friendly health services (AFHS), which are aimed at improving sustenance to long-term adherence to ART amongst APHIV [7]. The strategies, however, have proven to be ineffective as there is still evidence of poor sustenance to ART by adolescents. Poor adherence to ART by APHIV undermines efforts to stop the spread and increase in new HIV infections among adolescents and youth. Nabunya and colleagues argued for a need for transitioning from individual-based to family-focused health interventions, given the protective and supportive role of family in adherence to ART [20].

We undertook this study to investigate and contextualize the role of structural, familial, and psychosocial factors in sustaining long-term adherence to ART among APHIV. Understanding the pathway by which these factors may affect sustaining adherence to ART is critical to ensuring individual and public health benefits of ART. The findings will provide a point of reference for establishing adherence improvement strategies of APHIV in rural communities.

## 2. Materials and Methods

### 2.1. Study Design 

A mixed method, sequential exploratory design was employed to conduct one-on-one in-depth interviews in selected community health centres and clinics in the Vhembe district of Limpopo province in South Africa. This research design comprises two sets of individual and complete studies. In the first phase, qualitative data were collected and analyzed, and the results informed the development of a quantitative data collection tool. In this paper, the focus is on the findings of the qualitative strand using explorative design to explore the experiences, challenges, and support needs of APHIV. 

### 2.2. Study Setting

The study was conducted in three municipalities of the Vhembe district of Limpopo Province, South Africa. The Vhembe district has four districts and is largely a rural setting. The majority of its population are unemployed and live below the poverty line of R 1417.00 per month (USD 76). The unemployment rate is at 70%, with the youth as the worst affected. Small businesses and subsistence farming are the main economic activities, but the majority of citizens depend solely on supplementary grants from the Government [28].

The Vhembe district has 6 district hospitals, 1 regional hospital, 1 specialized hospital, 115 clinics, 8 community health centres (CHCs), and 19 mobile clinics that are used for outreach to the most remote areas. The study setting consisted of 16 facilities, which included 3 CHCs and 13 clinics which were randomly selected. All selected facilities offer ART to children and adults and depend on nurses who are trained to initiate and manage ART via the nurse-initiated management of antiretroviral therapy (NIMART). 

The standard distance for access to a hospital is a 5-kilometre (km) radius for a population of above 50,000; for clinics, it is 5 km for a population of above 10,000; and for mobile clinics, it is 7 km. However, in most communities, patients have to walk more than the standard radius of five km [28]. Those who cannot walk to the facilities use minibus taxis to access health care services.

### 2.3. Population and Sampling 

The population comprises 2137 APHIV from four municipalities in the Vhembe district. A purposive sampling technique was employed to select the 21 APHIV from the three municipalities that participated in the study.

Recruitment of APHIV took place telephonically and face-to-face with those who came to collect their medication. During telephonic recruitment, appointments were made for face-to-face meetings and one-on-one interviews. Purposeful sampling was used to select adolescents who were perinatally HIV infected and started ART before 10 years old. Some caregivers consented although the adolescents were not aware of their HIV status; this was discovered during interviews. The principal investigator made sure that questions which would have unintentionally disclosed their HIV status were avoided. Questions explored knowledge of the medication, what the medication was for, the duration of taking medication, and challenges they experienced with the medication. 

Adolescents above 18 years of age signed their own informed consent. For those below 18 years, assent was sought after caregivers provided a signed informed consent. The objectives of the study were explained before the signing of the consent form. All interviews were conducted on the same day after they finished their scheduled clinic consultation. A few opted to come on a different date and were provided transport fare for the interview. Refreshments were provided to all participants.

### 2.4. Data Collection Procedure

Qualitative data were collected using an in-depth interview guide. One-on-one interviews were conducted by the principal investigator with the assistance of two trained research assistants. The research assistants were trained in the processes of conducting interviews and the importance of maintaining confidentiality. An English interview guide was translated into Xitsonga and Tshivenda, respectively, and adolescents were interviewed in the language of their choice to afford them free expression of their experiences. Prior to each interview, written informed consent and assent were obtained after the interview procedures were explained, including the recording of the interviews using a digital recorder. In order to obtain detailed information, the interviewers probed accordingly and also used non-verbal communication like nodding, maintaining eye contact, and listening attentively. Interviews were conducted via a private consultation in the absence of the guardians to allow adolescents to speak freely. Interviews lasted a duration of 15 to 60 min, depending on the age of the participant and their knowledge of their HIV status. 

### 2.5. Data Analysis

Data were analysed using Tesch’s eight steps of qualitative data analysis [29]. Audio-recorded data were transcribed and translated into English for analysis. The researchers read through all the transcripts, jotting down ideas on the margins of each transcript in order to make sense of the information. The collected data were divided according to similar topics, which were then placed in columns of major topics, unique topics, and miscellaneous. The researchers went back to the transcripts with topics written in columns, and these were organized into codes. New codes that emerged were allocated to the data collected, and the best wording for each topic was identified. The total list of topics was reduced and converted into categories by grouping together the topics that related to each other. Each category was abbreviated and arranged in alphabetic order. Introductory analysis was conducted by collecting data material belonging to each category in one. Themes and sub-themes were summarised and sent to the independent coder. The final themes and sub-themes were independently identified by the researchers and the independent coder.

### 2.6. Ethical Approval 

The researchers obtained permission from the Turfloop Research Ethics Committee (TREC/228/2019: PG) and the Limpopo Department of Health Provincial and District offices. Informed consent was sought from APHIV aged 18–19 years old and from caregivers of APHIV aged 12–17 years old. Verbal assent was obtained from APHIV 12–17 years of age in the presence of their parents/caregivers. Participants were ensured of secrecy and confidentiality, and identifiers were removed during analysis to prevent unintended disclosure of their HIV status. 

## 3. Results

### 3.1. Description of the Study Sample

Twenty-one participants were interviewed. Their ages ranged from 10 to 19 years, with a mean age of 15.1 years. There were 17 girls and 4 boys. Sixteen participants were in secondary school. Eight of the adolescents were orphans under the care of grandmothers, siblings such as older sisters, aunts, and other relatives. Table 1 below illustrates socio-demographic characteristics of the participants.

### 3.2. Themes and Sub-Themes

Themes and sub-themes are presented in Table 2. Four themes emerged, which are food insecurity, psychosocial challenges, lifelong duration of ART, and inconsistent clinic attendance. Direct quotes from participants are presented in italics. Four themes and ten sub-themes emerged during data analysis, which are lack of food to sustain long-term adherence to ART, psychosocial challenges, inconsistent clinic attendance, and the needs of APHIV. 

#### 3.2.1. Lack of Food to Sustain Long-Term Adherence to ART

Lack or shortage of food emerged as one of the challenges leading to poor adherence to life-long medication that most participants encountered. Their narratives described how the death of their mothers, divorce, separation, and desertion affected their food security and subsequent adherence to medication. They often ran out of food and skipped their medication because they were told by healthcare professionals not to take their medication on an empty stomach.

They reported on factors that lead to food insecurity, such as living in severe poverty, orphanhood, divorce, or parental separation.

##### Orphanhood

Participant 009 also said “I am an orphan, my mother died and my father lives with another woman, he refuses to support me”. Whenever I get his contact details and call, he either blocks my number or changes it. I live on hand-outs from relatives and friends, at times I sleep on an empty stomach when no one gave me food and that is the only time where I do not take my medication”. 

Participant 013 further said “I do not get child support or foster grants, I depend on relatives and other people for food, both my parents passed away. At times I skip my medication because I get tired going to my relatives for food so I sleep hungry and you understand that I can’t take my medication on an empty stomach”.

##### Separation or Divorce

Participant 001 said “I stay with my father; my mother went back home. Since my mother left, my father refuses to buy me food, he only buys meat and other food when his girlfriend come to visit. Sometimes I skip my medication because I don’t want to take my pills on an empty stomach”.

##### Poverty

The data revealed that orphanhood, unemployment, and dependence on social grants (old age pensions, childcare grants, and foster care grants) subjected the adolescents to severe poverty, which further aggravated food insecurity and non-adherence. Once again, they expressed the fear of taking medication on an empty stomach as a reason for skipping ART doses. 

Participant 007 said ”I stay with three siblings, and I am the eldest. Both our parents died. I have to make sure that we get food every day and I am not working. I have a child who is getting child support grant of R350, which I use to buy him formula and diapers. I buy him low-cost diapers and formula for us to have money for food that is not enough. Sometimes I do laundry for people during weekends to get extra cash to sustain us until the month end, but it is not always easy, as I have to balance school and household chores and other things. Some days we run out of food, and it affects me too much because I sometimes skip my pills when there is no food since I can’t take my pills on an empty stomach”.

Participant 014 said “When my mother was still alive, she used to sell vegetables in the community to supplement the money that she got for childcare grant which is R350 for one child, so we were able to buy food to last the whole month. Now life is hard for me and my siblings since she died and we still depend on that R350 without any supplement, we live in severe poverty and sometimes it is hard to get money for food and other basics”.

Participant 005 said “I live with my grandmother, my mother works far in the farms, and she does not earn much, she sends money to buy food, but it is not enough to last us for a month, so the food runs out before she sends more money”.

##### Dependence on Social Grant

Dependence on social grants to sustain their livelihood increased the vulnerability of adolescents to severe poverty. Eleven of the adolescents reported that they were depending on the child support grant of R 350, which is an equivalent of USD 19.4. Two adolescents reported receiving a foster care grant of R 950 (USD 52.8), and six other orphaned adolescents were not receiving the foster grant due to different reasons. Social grants provided by the South African Government to combat poverty are not sufficient to sustain the livelihood of the recipient families and to buy sufficient food, as reflected in the quotes below. 

Participant 008 said “I live with my mother and my three siblings, and she is not working so we depend on social grant for food which is not enough to last us until the next pay day, so we sometimes have one meal a day for the food to last for a month which leaves us hungry most of the day, so it is not easy taking your medication daily”. 

Participant 005 said “there is no one working at home, we only depend on child support grant for food which is R700 for my two siblings. This money must buy everything in the house for our survival. My mother only buys 25 kg of mealie-meal and box of chicken feet sometimes with a box of fishes. This food does not last for a month since we are a family of seven people, and everyone must look for his/her food when there is no longer food in the house”.

Participant 007 said “I live with my grandmother, and she is the one who buys food with her grant, but the money is not sufficient because she has pays her stokvels and burial society with that money and also settle her depts. which leaves us with little money to buy food which becomes insufficient to last until the next pay day”.

Participant 011 said “I get foster care grant from social development and we use the money to buy food and other things like school uniform, clothes and pay community fares at home and we do not have other income because both our parents passed away and we are living on our own with my 2 siblings, so end up buying only the basic things like mealie-meal, stamp, chicken feet and sack of potatoes. The food lasts for only two weeks leaving us with two more weeks without enough food”.

#### 3.2.2. Psychosocial Challenges

The data revealed that secrecy and the fear of stigma emerged as the most common psychosocial challenges faced by adolescents to sustain adherence to lifelong ART and their day-to-day experiences of living with HIV. They reported being burdened by keeping HIV a secret from significant people in their lives.

##### Secrecy 

Society has normalized keeping one’s HIV diagnosis a secret from everyone except close family members. The adolescents reported that the secret of living with HIV was kept among close family members.

Participant 001 said “There is no one else who knows that I am taking this medication except my father and another lady who stays nearby because my father told me not to tell anyone because they will laugh at me”.

Participant 005 also said “I have been taking this medication from young age and it is only my mother and grandmother who knows because I was told that I must not tell anyone outside the family that I am taking this treatment, even the other family members do not know because they might tell other people about it”.

Participants 003, 004, and 007 said “no one know about the HIV status and that they were taking medication except their family members”.

##### Keeping HIV a Secret Is Burdensome

As stated in the preceding paragraph, adolescents kept their HIV status secret because society has normalized the secret nature of HIV. The data revealed that adolescents lived with the burden of living with a secret. They expressed that keeping their HIV status secret was weighing heavily on them. Their narratives are reflected below. 

Participant 011 said “living with HIV is traumatizing because it is a secret you cannot share with anyone. Ever since I was told that I am living with HIV, I feel like I am no longer the same person who lived a care-free life, now I have this secret that weighs on me, how do I become the same me again?”

Participant 015 said “I have a child with my boyfriend, but I did not tell him about my HIV status and sometimes I feel guilty because I know that I might have infected him also, but I can’t bring myself to tell him”. 

Participant 002 said “I am not free taking medication around my friends as they do not know that I am taking this medication, I must sneak out and take my medication when they are around. I feel tired keeping this secret, but do I have an option, no, I don’t, and it hurts because I did not do anything to live like this”.

Participant 016 said “sometimes I wonder who introduced HIV as a secret disease. We must live with this deep secret for the rest of our lives as if we did something wrong. Believe me keeping a secret weigh heavily on you because you are not free like other people, you must be always cautious”. 

##### Fear of Being Stigmatized

Stigma was the most reported psychosocial challenge, leading to poor ART adherence. The fear of stigma surrounding being HIV-positive was one of the motives for adolescents’ decision to keep their HIV status a secret. Furthermore, their narratives revealed that they feared being judged by significant others, such as their friends and other people in their communities. 

Participant 012 said “Living with HIV is so tough because you are scared that someone might find out about your HIV status, and they will alienate you. Sometimes when people talk about people living with HIV, (PLHIV), I feel uncomfortable because I can’t tell them the truth, but it is not nice. At times I want to talk to someone who will understand what I am going through, but who? You will just be giving people a topic to talk about”.

The fear of unintended disclosure and being judged affected adherence to clinic appointments and taking medication on time and consistently. 

Participant 017 said “I am not free collecting my medication during the week as I am afraid that I might bump into someone I know, and they will know that I have HIV which will not be nice”.

The fear of perceived stigma and discrimination surrounding HIV diagnosis deterred adolescents from disclosing their HIV status to friends. They did not want anyone outside their family members to find out about their HIV status for fear of being discriminated against and being labelled as promiscuous.

Participant 005 said “I would not want my friend to know my HIV status because HIV is “that illness, it is not like any other disease, it is a disease which cannot be cured, and people look at you as if you are not human enough when you have it”. 

Participant 001 said “I have never been laughed at, but they do laugh at people who are HIV positive”.

Participant 006 said “My grandmother told me that I must take my medication for HIV so that I don’t get sick and ends up dead like my mother did, but I do not want to have HIV because it’s for older people who sleeps around, and people laugh at you when you have it”.

#### 3.2.3. Structural Challenges

The data in this study found that there were structural challenges faced by APHIV which deterred them from adhering to ART. They failed to collect scheduled medication due to clashes with school activities. Furthermore, they were unable to access health facilities with ease due to a lack of money for transport. As mentioned above, APHIV were living in families with severe poverty; hence, they could not afford transport to collect their medication.

Participant 002 said “there are times when I fail to go to the clinic to collect my medication, not because I don’t want to, but I am staying far from the clinic, and I have to catch a taxi and there is no money’. My aunt used to accompany me to collect medication, but she is now in her boyfriend’s house, so I don’t have means to get to the clinic to fetch my medication”.

Participant 004 said “I have asked nurses at the clinic to schedule my dates on weekends since I knock off late at school and they refused citing that they are understaffed on weekends and cannot afford chronic medication. Consequently, I fail to fetch my medication on time because I do not want to miss out on important things such as tests or examinations. I don’t want my teachers to suspect that I am taking medication, so I don’t ask them for permission to go to the clinic”.

#### 3.2.4. Needs

During the interviews, the adolescents indicated that they desired to have someone to talk to when they felt overwhelmed living with HIV and having to take lifelong medication. While most of them were on the once-daily combination pill, they still wished that there could be a modification of the regimen. 

##### Need for Mentors and Supporters

The adolescents expressed the desire to have a mentor who would ‘always’ be available for them. Some of them pointed out that it was not easy for them to ask the nurses some of the questions they had regarding HIV and living with HIV positively.

Participant 002 said “Sometimes I feel like talking to someone and you do not know who to talk to, I was wondering if it is possible that we can have a mentor who will always be available in case we have questions or uncertainties, someone we can call even after hours”.

Participant 014 said “I can say I have a good relationship with the nurses, but I am not used to them (nurses) as I am a shy person. I have questions that I would like to ask them sometimes, but I am afraid to ask. Like I want to know as to why I am taking treatment daily, but I am not sure who to direct these questions to”. 

##### Treatment Modification

Some APHIV suggested that if their medication could be modified to smaller tablets, it might ease their lives and improve sustenance adherence to ART.

Participant 001 said “sometimes I don’t take my medication because some of these pills are too big for me to swallow. I want the big pills changed, because they choke me”. 

Participant 009 added “This medication is difficult to take because of the strict schedule and the frequency, I feel like it would be better if the government could consider changing these pills to an injectable or make treatment schedule to be flexible, it would make our lives easier”.

## 4. Discussion

The study explored the familial, structural, and psychosocial factors that affect long-term adherence to ART by APHIV. The sample comprised APHIV who were informed of their HIV status in their late teens. We found that although APHIV showed interest in adhering to treatment, they faced multifactorial challenges that resulted in poor long-term adherence to ART. 

Fear of stigma was a major challenge that contributed to APHIV keeping their HIV status a secret. Previous studies reported on how stigma and prejudice against HIV influenced APHIV to keep their HIV status a secret [13,30,31,32]. Secrecy affected their adherence to ART as they missed taking their medication in the presence of friends and other people [4]. Furthermore, the fear of stigma affected clinic attendance as some of them skipped their clinic appointment to avoid being seen collecting their medication. Other studies reported similar findings [14,15,30]. Similar to previous studies, APHIV felt burdened by keeping their HIV status secret, which deprived them of emotional and psychosocial support from significant others [3,10,33]. Interventions that include social support, secrecy and disclosure counselling, and comprehensive ART education are effective in improving adherence and are acceptable to APHIV [4].

Consistent with previous studies conducted in South Africa and other SSA countries, most APHIV live in socially depressed environments and are subject to extreme poverty, characterized by frequent hunger [17,34]. In these settings, most households are food insecure, and incomes are primarily constituted by social grants [18,33]. In the current study, APHIV indicated that frequent hunger complicates their lives as they find it difficult to take ART on empty stomachs. Food insecurity increases the risks of poor nutritional status in PLHIV infection, thereby compromising the effectiveness of ART and sustenance [21,26].

We further found that in this study, poverty and food insecurity were exaggerated by orphanhood. Six APHIV were maternal orphans, and two were double orphans. Running out of food in the middle of the month was a common occurrence in most households. While subsistence farming was a means to produce food and reduce food shortages, this did not continue after the death of parents. These findings concur with the results of other studies. HIV infection causes food insecurity by reducing agricultural production, reducing income, and increasing medical expenses, leading to a reduced capacity to procure or produce food [26,35,36]. 

The study found that clashes between school activities and clinic appointments affected ART adherence. In line with other studies, APHIV skipped their clinic appointments when they clashed with important school activities such as tests and exams [14,37,38]. Additionally, APHIV often missed clinic appointments due to fear of being questioned if they sought permission to miss school. As already stated, APHIV in this study lived in socially deprived environments; thus, transport was a structural barrier to accessing health facilities for routine ART refill [3,13,14,39]. The rurality of the setting meant that the clinics were far and because of the secrecy about their HIV status, the adolescents could not ask a friend to accompany them out of concern that it would lead to unintended disclosure. Similarly, other studies reported difficulties in accessing health facilities as a key structural challenge for APHIV [13,14].

Furthermore, lack of adolescent-friendly health services and lack of emotional support from the nursing staff were barriers to linkage to care, which affected APHIV’s capacity to adhere to ART. The findings acknowledge the gap in emotional support, which could be due to staff incompetence in managing APHIV. Consistently, other studies proposed an upgrade of healthcare workers’ skills in managing APHIV as an evolving population. There is a need for the integration of youth-friendly services within ART programmes. Literature advocates for after-school clinics, adolescent clinic days, teen clubs, peer support groups, and youth-friendly waiting areas [5,13,40].

## 5. Conclusions

We found poor adherence to long-term ART among APHIV. Most of the barriers to adherence in this study were beyond their control but attributed to structural and familial factors. The findings suggest that poverty influenced APHIV sustaining adherence to long-term ART. Living in poverty-stricken households affected their ability to honour clinic appointments and have adequate food supply to facilitate taking their medication consistently. 

In addition to institution-based interventions, there is a need for family, community-based, and multi-sectorial interventions to support long-term adherence among APHIV. The role of social and financial support from the family in influencing adherence outcomes has been well established in research. However, in the case where APHIV are orphaned or live in disorganized households, interventions that involve services offered by the Department of Social Development, such as the provision of food parcels to alleviate and address food insecurity, are crucial for optimal adherence to medication. Furthermore, for APHIV living in socially depressed environments, a supportive environment provided via the integration of youth-friendly services within ART programmes will play a significant role in improving adherence and providing the mentors desired by the adolescents. 

Inconsistent adherence to long-term ART observed in the current study might be the result of a lack of involvement of adolescents in the design of interventions that target them. It is imperative that prioritized interventions to improve ART adherence in adolescents are context-specific, culturally appropriate, and respond to the identified needs of adolescents. Strategies can only be effective when there is increased knowledge of the factors impacting ART adherence.

The study presents the findings in the rural context and provides a basis for further research on the improvement of APHIV’s lives.

## Figures and Tables

**Table 1 ijerph-20-07074-t001:** Sociodemographic characteristics of respondents.

Place of stay	Rural	12
	Semi-urban	5
	Urban	4
Education level	Primary	5
	Secondary	16
Years on ART	Under 5 years	0
	Above 5 years	21
Years of orphanhood	Under 5 years	2
	Above 5 years	6
Disclosure status	Disclosed	18
	Not disclosed	3
Who do they live with	Both parents	3
	One parent	10
	Guardians/grandparents	8

**Table 2 ijerph-20-07074-t002:** Summary of themes and sub-themes.

Main Theme	Sub-Theme
Lack of food to sustain long life ART	Orphanhood Separation/divorce/desertion Poverty Dependency on social grant
Psychosocial challenges	Secrecy Keeping the secret is burdensome Fear of being stigmatized
Structural challenges	Lack of transport money to the clinic Clash of clinic appointments with school schedules
Needs	Need for mentors and support Need for treatment modification

## Data Availability

The data supporting the results are available from the corresponding author, R.A.M., upon reasonable request.

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
