# Peer review of "Structural, Familial, and Psychosocial Factors Affecting Long-Term Antiretroviral Treatment Adherence amongst Adolescents Living with Perinatally Acquired HIV in Limpopo, South Africa"

_ijerph, 2023, doi:10.3390/ijerph20227074_

Round 1

Reviewer 1 Report

Comments and Suggestions for Authors

1) In abstract, Line 13 first use full form than abbreviation

2) Line 17, there is little data available on 16 factors such familial and structural?? sentence incomplete.

3) Line 36 and line 13 is same? it must be change.

4) line 39 the same level of improvement? what level please describe?

5)  one-pill, once-daily, fixed-dose combination antiretroviral regimens? which one?

6) first discuss this According to WHO sustaining 57 adherence means attending 80% of the clinic appointments than studies? in paper first you discuss studies than definition.

7) Method section needs discussion regarding the data collection. its very complex writes. I suggest to re write like a way that readers have no idea about the topic.

8) Results section needs more improvement. Results must be in tabular form and graphical forms for more understanding.

Author Response

we thank the reviewer for reviewing our manuscript

Reviewer 2 Report

Comments and Suggestions for Authors

The article on the "Structural, familial and psychosocial factors that affects long term sustenance on ART amongs adolescents with perinatal HIV in rural setting of South Africa" by Mabasa, Madiba and Mothiba is an interesting and important piece of work.

* The methodology is explicitly explained and scientifically sound. 

* The methods and results are detailed, providing long quotes that give a very accurate picture of the extent of the challenges that APHIV face.

* The sentence structure & grammar however needs a lot of improvement.  Currently, overall, the manuscript it is too wordy or verbose and could do with cutting down.

* The sections also need some re-structuring so that results do not appear in the methodology and the discussion does not re-state methodology, and results too much or include information that should be in the introduction/background session. 

* Specific comments about English language are provided in the proceeding section but I have attached a pdf scan to this section where I have written notes on, concerning the first two pages of the manuscript.  These should give the authors an indication about the types of changes they need to make to the rest of the manuscript.  

Some general comments that are not related to English Language are: - 

1.  Line 40:  Is this comment refering to a lack of improvement in adolescents world-wide or Sub-saharan African or South Africa in particular?  Please specify throughout when you are focusing mainly on Africa, as your references seem to indicate, rather than the whole world.

2. Line 43: as above, specify which region and only cite the specific reference.  i.e., does this comment refer to the Ugandan study specifically?

3. Lines 52 to 54:  Again, this is a very generalized statement, do you mean in Africa or South Africa.  "Majority", based on which data source?  Can you quantify this more specifically, i.e., number of studies published recently or percentage range based on references found on the topic for a specific region?

4: Lines 117 to 124 within section 2.2:  You are mixing results into methodology here.  Just outline the methods.  Numbers of adolescents who were aware of their HIV status should come in the results.  There is also too much explanation about exactly what happened during the telephone conversations.  Cut this down to say participants were recruited by telephone.  You could say that the initial 10? telephone conversations led to the unintentional disclosure of HIV status and question XYZ was therefore altered as indicated (in supplementary section??) to ensure this was avoided in subsequent calls.  

5.  Detailed questionnaires and any other tools used during the study should be included in supplementary material or provided in a cloud-based data bank such as the osf portal - https://osf.io/.

6. Is there a reference that details the "Tesch eight steps" of qualitative data analysis?  This should be cited everytime it is stated.

7. The results are very detailed.  The detailed quotes are very insightful.  Summary of the results for each section should be provided after the detailed accounts so that it is clear where the summaries are derived.  Currently they are included within the conclusions.  You could consider truncating the quotes within the main manuscript and providing detailed quotes from the participants in a supplementary section...

8. Generally conclusions come before the discussion and are the logical conclusion based on the results.  I would prefer to see the conclusions before the results, unless this is the manuscript structure prefered by the journal.

9. The conclusions are too verbose and unstructured.  Currently they include too much methodology, background and results.  The discussion should put the results and conclusion into the context of current literature and expound on them further and give recommendations for the future.  They are an expansion or exposition of the conclusions and should not really re-iterate background or results.  They should not include information that should have already been in the introduction or background.  They can re-state results in as far as is needed to give context to a segment of the discussion.  I suggest the discussion should perhaps focus on why the challenges indicated in the manuscript were not really known or taken into account when the current interventions for adolscents were designed.   You can expand on the details of the current interventions and the history f their implementation or exactly what support is currently provided and why.  The authors should highlight what this study has unearthed that is new or more concrete compared to previous studies with references to these previous studies.  The discussion should expound on the conclusions and provide future recommendations  with contextual detail.  The authors could discuss/recommend how exactly can further support potentially be provided to these adolescents in each of the areas that are needed i.e. familial, structural and psychosocial - exactly what extra support do the authors suggest.  They could give more context to the extent of poverty and orphan hood in limpompo province with statistics, duration and how this has been compounded by HIV.  What is the prevalence and incidence of HIV in this region and how does it compare to other regions in South Africa.  Is it typical to South Africa or is this a particular hot-spot of the pandemic? What interventions have been tried in the past and how successful have they been and if not why?  How exactly can community groups be facilitated to improve the situation for these adolescents for example, and what could be done to improve their success?  Authors could perhaps cite examples from other, similarly affected-regions, where more-successful interventions have been applied.  Did the questionnaires or interviews discuss with the adolescents any specific measures that would help them?  Could they be provided with more food-supplementation perhaps, apart from the current social-care funds provided by the government?  Could a bi-weekly supply of nutritional drinks/supplements work for them or vouchers to food-banks?  Could community centres that provided meals be set up with vouchers for these youth, to ensure they get regular meals?  Some more detailed practical recommendations would be useful.

Comments on the Quality of English Language

*The sentence structure and grammar needs thorough editing and revision.  Some examples are provided below to give an indication of the level of revision that is required in this area: - 

1.  The title would read better as "Structural, familial and psychosocial factors that affect long-term ART-adherence amongst adolescents living with perinatal HIV in the Limpompo Province of South Africa".

2. The phrase or variations of the phrase "sustenance of long term adherence to ART" should be replaced with "long-term ART-adherence" or something similarly succinct throughout the manuscript.

3.  Line 14 would read better as something like this: "...in sub Saharan Africa, there is still poor long-term ART-adherence due to multiple factors, with the consequence of..."

4.  Lines 15 to 16:  If you include the "multifactorial challenges" in the previous sentence before as suggested in Line 3,  the sentence starting "They are faced with multifactorial challenges..."  can be cut out.

5. As indicated in the image attached to the previous section, the Abstract could be edited and clarified as follows(Lines 16 to 32): - "There is little data available on the familial and structural factors which affect long-term ART-adherenceMethods: A qualitative, exploratory  design was used to conduct in-depth interviews with 21 APHIV attending HIV care and management in ... Limpompo District, South Africa. Transcipts... into English... Findings: The samples... 10-19 year old APHIV...(Instead of "Most" Provide a specific advantage or number e.g., All) of them had received ART for more than 5 years... extended, disrupted, granparent- and child-headed households... food insecurities due to poverty, orphan hood.... households, which deterred them from long-term ART-adherence.  In addition, dependence on social support grants... APHIV had challenges ... clinical attendance, ... clinical appointments and the lack of ... to the clinic, which affected ART-adherenceConclusion: Although APHIV were on one-pill, fixed-dose ART, there was poor long-term adherence to it due to familial, structural and pyschosocial challenges.  ... there is a need for family, commmunity-based and multi-sectoral interventions to support long-term ART-adherence among APHIV."

6. I the image attached to the previous section, I have attached a scanned copy of my hand-written edits to lines 35 to 93.

7. Similar editorial work is required for the rest of the document.

Author Response

We thank the reviewer for their comments.

Round 2

Reviewer 1 Report

Comments and Suggestions for Authors

Revised version is accordingly in terms of scientific.  All changes are accepted. 

Author Response

Thank you for reviewing our work. see attached document.

Reviewer 2 Report

Comments and Suggestions for Authors

This is a significant improvement from the last version of the manuscript.

The narrative is much clearer and more logical.  The problem / issues are clear from the data and the recommendations make sense.

The conclusion and discussions are sound.

One specific descrepancy notices was: - 

The paragraph starting at line 215 versus the one starting at line 219.  Both refer to "participant 009" but in the first paragraph this participant is an orphan with a mother who passed away and a father who lives with a new wife while in the latter paragraph, "participant 009" has both parents who have passed away and has different circumstances affecting their ability to adhere to ART.  Please check your records to determine whether or not you are refering to the same person and correct the details accordingly.

Comments on the Quality of English Language

The English is much improved. 

There is still quite a bit of unnecessary repetition, e.g., lines 386 to 399, repeats the same sentence and theme over an over again - i.e. secrecy and stigma found in other studies and recapitulated in this one.  These 4 to 5 sentences, could be summarized in one all-encompassing sentence.  The authors should go through each paragraph critically and see if they are repeating themselves too much and determine where sentences could be combined and/or eliminated.

Unlike before, however, the repetition does not detract from the main message too much.

Here are a few other minor corrections I noticed: - 

Line 13: "... was used to conducted in-depth interviews..."

Line 15: "... were translated  into English...!

Line 18: "... grandparent- and child-headed households,..."

Line 19: "... orphanhood and due to living in ..."

Line 20: "... deterred them from long-term adherence..."

Line 21, 24, 26 and elsewhere: "... long-term..."

(https://www.techtarget.com/whatis/feature/Which-is-correct-long-term-or-long-term)

Line 23: "... one-pill fixed-dose ART..."

Line 37: "... fail to cope with living with HIV,..."

Line 61: "... and medication-related factors..."

Line 62: "... , which affected adherence to treatment, have also..."

Line 116: "... affected. Small businesses..."

Line 117: "... majority of citizens depend solely on supplementary grants..."

Line 139: "...started ART before 10 years old..."

Line 149: "... their scheduled clinic..."

Line 296: "...because society has normalized..."

Line 297: "... keeping their HIV status..."

Line 389: "...APHIV to keep their status a secret..."

Please go through the rest of the manuscript with a fresh pair of eyes and try catch as many of these minor errors as possible before final submission.

Thank You.

Author Response

Thank you for reviewing our manuscript. See attached document 
